# Cellular and Synaptic Dysfunctions in Parkinson’s Disease: Stepping Out of the Striatum

**DOI:** 10.3390/cells8091005

**Published:** 2019-08-29

**Authors:** Nicolas Mallet, Lorena Delgado, Marine Chazalon, Cristina Miguelez, Jérôme Baufreton

**Affiliations:** 1Université de Bordeaux, Institut des Maladies Neurodégénératives, F-33000 Bordeaux, France; 2CNRS UMR 5293, Institut des Maladies Neurodégénératives, F-33000 Bordeaux, France; 3Department of Pharmacology, University of the Basque Country (UPV/EHU), 48940 Leioa, Spain; 4Laboratory of Neurophysiology, ULB Neuroscience Institute, Université Libre de Bruxelles, 1070 Brussels, Belgium

**Keywords:** Globus pallidus, subthalamic nucleus, substantia nigra, dopamine, pacemaking, neuronal excitability, GABAergic transmission, neuronal oscillations

## Abstract

The basal ganglia (BG) are a collection of interconnected subcortical nuclei that participate in a great variety of functions, ranging from motor programming and execution to procedural learning, cognition, and emotions. This network is also the region primarily affected by the degeneration of midbrain dopaminergic neurons localized in the *substantia nigra pars compacta* (SNc). This degeneration causes cellular and synaptic dysfunctions in the BG network, which are responsible for the appearance of the motor symptoms of Parkinson’s disease. Dopamine (DA) modulation and the consequences of its loss on the striatal microcircuit have been extensively studied, and because of the discrete nature of DA innervation of other BG nuclei, its action outside the striatum has been considered negligible. However, there is a growing body of evidence supporting functional extrastriatal DA modulation of both cellular excitability and synaptic transmission. In this review, the functional relevance of DA modulation outside the striatum in both normal and pathological conditions will be discussed.

## 1. Introduction

The basal ganglia (BG) participate in a great variety of functions including motor programming and execution, procedural learning, cognition, and emotions [1,2]. Dopaminergic innervation provided by mesencephalic dopamine (DA) neurons from the *substantia nigra pars compacta* (SNc) plays an essential role in the control of BG functions by modulating cellular and synaptic properties at each stage of the BG network. DA modulation of the striatal microcircuit has been extensively studied over the past decades and has been instrumental to better understand striatal function in health and disease states (for reviews, see [3,4]). The degeneration of midbrain DA neurons and the subsequent loss of DA in the BG trigger cellular and synaptic alterations, which are believed to be responsible for the appearance of the motor symptoms of Parkinson’s disease (PD). Dopaminergic innervation of the striatum (STR) is acknowledged to be by far denser than in the rest of the BG. Nevertheless, there is a growing body of evidences supporting the existence of discrete, functional dopaminergic innervation of extra-striatal nuclei (ESN), namely the subthalamic nucleus (STN), the internal and external segments of the globus pallidus (GPi and GPe, respectively) and the *substantia nigra pars reticulata* (SNr) [5]. In line with this evidence, the action exerted by DA on ESN has been investigated only recently and is the focus of the present review. After a brief description of BG network organization, we will discuss recent discoveries supporting a functional role of DA outside the striatum in both healthy and disease conditions.

The STR is the main entry structure of the BG and the principal recipient of cortical inputs. This GABAergic nucleus consists of 95% projection neurons, recognizable by their small soma and their dendrites covered by spines—therefore called spiny projection neurons (SPN). The remaining 5% of the STR is composed of several classes of GABAergic and cholinergic interneurons [6,7]. SPNs give rise to two pathways that propagate cortical information directly or indirectly to the output of the BG. SPNs of the direct pathway (dSPNs) express dopamine D1 receptors (D1Rs) and primarily innervate BG output neurons of the SNr and GPi (called the entopeduncular nucleus in rodents, EPN) in a monosynaptic manner. On the other hand, SPNs of the indirect pathway (iSPNs) express dopamine D2 receptors (D2Rs) and project to BG outputs via a polysynaptic route involving the GPe (called the globus pallidus in rodents, GP) and the STN. As predicted by the anatomical-functional model of the BG [8], it has been shown experimentally that these two pathways exert opposite control over motor execution [9] (Figure 1). However, this view of opposite functions of the direct and the indirect pathways has been challenged by a study showing that both iSPNs and dSPNs are co-activated before movement initiation [10]. This coordinated activity has been interpreted as dSPN activation representing a process of selection/activation of a desired movement, while simultaneously activating iSPNs provides global suppression of alternative/unwanted actions [11]. In addition to these two pathways, multiple loops within the BG network, such as the hyperdirect cortico-subthalamic pathway [12], the reciprocally-connected GP-STN loop [13], the pallido-striatal pathway [14,15], and bridging collaterals from dSPNs in the GP [16,17], certainly contribute to normal and pathological operations of the BG network (Figure 1).

The aim and scope of the present review are to provide a comprehensive overview of the current literature regarding the role of DA in the modulation of ESN activity in both healthy conditions and in PD.

## 2. Extra-Striatal Nuclei Neurons Share Specific Features

### 2.1. ESN Neurons are Fast-Spiking Autonomous Pacemakers

A striking feature of ESN neurons compared to SPN is their ability to fire action potentials (APs) in a completely autonomous manner (i.e., in absence of excitatory synaptic inputs). While several voltage-dependent and voltage-independent conductances contribute to maintaining the membrane potential of ESN neurons depolarized, persistent sodium and Kv3-family potassium channels appear to play a critical role in the capability of GP, STN, EPN, and SNr neurons to perpetually fire APs at a high rate [18,19,20,21,22,23]. Because of their very hyperpolarized potential [24,25], SPNs need to receive convergent excitatory inputs to fire APs and convey information downwards through the BG network [26,27,28,29]. In contrast, because ESN are autonomous pacemakers, their spiking activity is instead independent of excitatory inputs [30]. Therefore, information encoding and transfer by these neurons relies more on changes in their firing rate and pattern rather than on spiking activity *per se*.

### 2.2. GABAergic Transmission Efficiently Sculpts the Activity of ESN Neurons

Another interesting advantage of pacemaking for neurons is the emerging properties related to AP timing relative to incoming afferent inputs [31,32,33]. Most of the ESN except for the STN, the only glutamatergic nucleus of the network, are inhibitory nuclei. Therefore, GABAergic transmission, through the activation of ionotropic GABA_A_ and metabotropic GABA_B_ receptors, is certainly the most efficient system to control the firing of neurons and induces profound changes in their firing rate and pattern [34,35]. Hyperpolarization of the membrane potential resets ESN neuron pacemaking activity and engages voltage-gated channels. Activation or deactivation of those channels will transiently modify the intrinsic excitability of ESN neurons following the inhibition [21,36,37,38], leading to complex sequences of activity. These interactions between synaptic inhibitory inputs and voltage-gated channels can contribute to the emergence of transiently correlated and synchronous activity within and among the different nuclei, both in healthy and pathological states (see Section 7).

### 2.3. ESN Receive Functional Dopaminergic Innervation

In the brain, the STR is by far the principal recipient of dopaminergic inputs arising from the SNc [39]. The dopaminergic modulation of the striatal microcircuit has been well characterized and is the subject of many reviews [40,41], so it will not be covered here. Even if, in comparison, dopaminergic innervation of the ESN can appear negligible compared to the STR, there are several lines of evidence supporting functional extra-striatal dopaminergic innervation. Dopaminergic fibers and synapses have been detected in the GP and the STN using light and electron microscopy [42,43,44,45]. DA release has also been measured in the STN upon electrical stimulation of dopaminergic inputs using fast-scan voltammetry [43]. In addition, DA receptors are expressed in all ESN (for a review, see [46]), and local applications of dopaminergic receptor agonists and antagonists locally in various ESN affect the neuronal activity of these nuclei [47,48,49,50]. All these data suggest that DA participates in the normal function of the ESN by regulating both cellular excitability and synaptic transmission.

## 3. The Globus Pallidus

### 3.1. Neuronal Diversity in the GP

The GP has long been considered as a simple relay nucleus in the indirect pathway of the BG despite anatomical/immunohistochemical and functional/electrophysiological studies suggesting a certain degree of diversity in this nucleus [51,52,53,54,55,56,57]. More recent investigations have revealed the molecular [58,59] and functional complexity [60,61,62] of the GP. Several cell types have been identified based on the expression of specific molecular markers and electrophysiological properties [63,64,65,66]. Even if a unified nomenclature is still missing [67], two main clusters of neurons are emerging from single-cell RNA profiling [59]. The first cluster is composed of the so-called prototypic GP neurons which project to downstream nuclei (the STN, the SNr, and the EPN) but also to the STR [63,68,69]. These neurons express the transcription factor (TF) Nkx2.1 and show a regular fast-spiking discharge ex vivo [63,64]. This population represents more than 70% of the total number of neurons in the mouse and the rat GP and include of a subgroup, representing approximately ¾ of prototypic GP neurons, which also express the calcium binding protein parvalbumin (PV) in addition to Nkx2.1 [63,70]. The second cluster is made of arkypallidal GP neurons, which were described in the seminal study of Mallet and colleagues [14]. This cell type, which represents between 15 to 28% of GP neurons depending on the species studied [63,70], is characterized by the selective expression of the TF FoxP2 and a low and irregular firing discharge. The main distinguishing feature of arkypallidal neurons is their exclusive and massive axonal projections to the STR, which certainly constitutes the main external source of GABA of this nucleus [14]. It is interesting to note here, that all arkypallidal neurons co-express the TF NPas-1, but this marker is not specific to this neuronal population, as it is also found in a small proportion of prototypic GP neurons [63,65,70].

The activity of prototypic and arkypallidal neurons has been characterized in vivo under anesthesia both in the DA-intact (DI) and the DA-depleted (DD) rat [63,71]. In anesthetized rodents, prototypic neurons are the population most affected by DD, with the appearance of pauses in their activity leading to a reduced mean firing rate compared to the DI rat [63]. Several studies suggest that the emergence of these pauses is due to an exaggerated drive from iSPNs [72,73,74], maybe because iSPN preferentially send projections to prototypic neurons [75]. Together, these observations fit with the hypoactivity of GP neurons predicted by the anatomo-functional model of the BG [8]. In contrast, arkypallidal neuron activity seems less affected by DD, as only their phase-coupling increases without significantly changing their firing rate or pattern [63]. These alterations in GP neuron activity in vivo can be due either to changes in the intrinsic excitability and/or to changes in their afferent synaptic input drive (discussed in Section 3.4 and Section 3.5).

### 3.2. Ionic Conductances Underlying Pacemaking in GP Neurons

GP neurons exhibit a large range of firing activity and rates (between 0 to 100 Hz) in rodents and non-human primates in vivo [57,76,77,78]. Recordings of GP neurons in dissociated neuron cultures and in brain slices have demonstrated that this variability in firing is correlated with the type and density of voltage-gated conductances expressed by individual GP neurons [53,79]. Despite these individual differences, a subset of voltage-gated channels plays a key role in the autonomous pacemaking of GP neurons. Repetitive emission of APs by GP neurons relies primarily on persistent sodium and hyperpolarization-activated, cyclic nucleotide-gated cation (HCN) channels [21,80], which depolarize the membrane potential at subthreshold potentials. The regularity and precision of GP neuron firing is governed by both calcium-activated SK channels and HCN channels [81], while Kv3-family potassium channels promote spike repolarization in these neurons, enabling high frequencies of discharges [22].

### 3.3. Dopamine Modulation of Intrinsic Excitability of GP Neurons

DA modulation of the excitability of GP neurons has been suggested by various techniques, showing either the presence of dopaminergic fibers and receptors in the nucleus [42,44,82], the release of DA [83,84,85], dopaminergic modulation of voltage-gated ion channels [86], and neuronal excitability in vivo [87]. Interestingly, D2 receptor mRNA is present in all classes of pallidal neurons, with the pallido-striatal neurons expressing higher levels of D2 transcripts than pallido-subthalamic cells [54,55]. This suggests a differential modulation by DA of prototypic and arkypallidal GP neurons. Thus, it is likely that DD preferentially affects the intrinsic excitability of one population of GP neurons. DA modulation of GP neuron excitability is poorly understood, and so far, only one study has shown a direct action of DA on GP neuron voltage-gated conductance. In this study, the authors showed that D2Rs inhibit Ca_V_2.2 (N-type) channels in a protein kinase C (PKC)-dependent manner [86]. This result suggests that D2R activation reduces GP neuron excitability (but see Section 4.1).

Moreover, some GP neurons lose their autonomous pacemaking after DD [88]. The mechanism underlying GP neuron pacemaking loss involves a Ca_V_1.3 (L-type) calcium channel-dependent downregulation of HCN2 channels [88]. It also seems that this alteration of excitability of GP neurons preferentially affects NPas1-expressing pallido-striatal neurons. Indeed, the firing rate of the main subpopulation of prototypic neurons, PV+ neurons, remains unaffected in DD mice [89]. As a consequence, these results suggest that the change in the pattern of activity of PV+ GP neurons observed in vivo [63,71] relies on changes in the activity of iSPNs. Indeed, the hyperactivity of iSPNs promotes pauses in activity of prototypic cells [72,73,74]. It is interesting to note that, while excitability of Npas1 GP neurons is reduced, the pallido-striatal GABAergic inhibition provided by these neurons is increased in all subtypes of striatal neurons [89], suggesting the existence of a compensatory mechanism.

### 3.4. Dopamine Modulation of Gabaergic Transmission in the GP

GP neurons receive two main sources of GABA: extrinsic inputs coming from iSPNs and some collaterals of dSPNs [16,17,90] and intrinsic inputs from local axon collaterals [91]. Striato-pallidal inputs are mainly localized on proximal and distal dendrites of GP neurons [92], while pallido-pallidal inhibitory synapses are found on the soma and proximal dendrites of GP neurons [91]. Regarding their properties, striato-pallidal synapses are characterized by short-term facilitation (STF) [93,94] and their strength is modulated by a plethora of G protein–coupled receptors (GPCR) [95,96,97,98,99,100,101,102] including presynaptic D2Rs, which decrease the probability of GABA release [93,103]. In contrast, pallido-pallidal synapses are characterized by short-term depression (STD) [93,94,104] and are not modulated by presynaptic D2-like receptors [93]. GABAergic transmission is also regulated by postsynaptic D4Rs, which reduce the amplitude of GABA_A_-mediated current through the suppression of protein kinase A (PKA) activity [105]. The paired-pulse facilitation observed at striato-pallidal connection suggests that this synapse has a low initial release probability. However, a full characterization of unitary iSPN-GP connections will be required to better understand the release dynamics of these synapses. On the other hand, paired-recordings of GP neurons have revealed that despite the sparse connectivity (~1%; [91,104]) and STD, unitary pallido-pallidal transmission is able to reduce the postsynaptic firing rate through a combination of chloride driving force, synaptic summation, and incomplete STD [104]. The sparse connectivity combined with the relative efficacy of axon local collaterals certainly contributes to uncorrelated activity of GP neurons recorded in vivo under healthy conditions [78].

### 3.5. Astrocyte-Dependent Alteration of Gabaergic Inhibition in the GP

The GP is the nucleus in the BG with the highest astrocyte density [106], suggesting an important role of these glial cells in regulating GP motor function. Indeed, voluntary exercise in mice triggers astrocyte structural plasticity. This consists of elaboration of perisynaptic astrocyte processes (PAP) [107], which are dynamic elements thought to regulate synaptic transmission by clearing neurotransmitter and releasing gliotransmitter into the synaptic clefts [108,109,110]. GP astrocytes express GAT-3 GABA transporters [111,112], which are primarily localized on PAPs [107]. They also express a variety of ionotropic and GPCR receptors including DARs, suggesting that DA somehow modulates intracellular signaling. Calcium imaging experiments have shown that D3R activation triggers a reduction in intracellular Ca^2+^ waves under control conditions, while DD exerts the opposite effect [113]. We also observed that DD is responsible for GP astrocytosis [106], which suggests astrocyte dysfunction in PD. Two recent studies have elucidated distinct astrocyte signaling pathway impairments that promote an increase in GABAergic transmission in the GP. In the first study, the authors demonstrated that after DD, glutamate release from GP astrocyte is reduced, which dampens the activity of pre-synaptic mGluR3 and in turn increases GABA release at striato-pallidal synapses (Figure 2) [113]. A second study showed that DD also induces a reduction of the expression of GAT-3 in GP astrocytes, leading to reduced uptake and elevation of ambient extracellular GABA levels promoting the activation of extrasynaptic GABA_A_R-mediated tonic inhibition (Figure 2) [114]. Both astrocyte glutamate release and GAT-3 uptake activity seem to be regulated by D2-like family receptors, suggesting that reduced stimulation of these receptors in DD conditions is responsible for the increase of extracellular GABA concentrations [115] and GABAergic transmission in GP neurons. Overall, GP neuron hypoactivity in vivo seems to be the consequence of several convergent mechanisms—at the circuit level, with the hyperactivity of iSPNs [72,73], but also locally, with the reduction of GP excitability and the dysregulation of GABAergic synaptic and extrasynaptic transmission [92,113,114] (Figure 4). Restoring autonomous pacemaking in the GP by overexpression of HCN2 fails to re-establish motor function in DD animals [88]. Interestingly, because GP astrocytes play a key role in the regulation of GABAergic transmission, restoring normal levels of inhibition in the GP by manipulating astrocyte function appears to be an attractive and promising therapeutic strategy to reduce PD motor symptoms.

## 4. The Subthalamic Nucleus

### 4.1. Dopamine Modulation of STN Neuron Excitability

Because the change in the firing pattern of STN neurons is considered an electrophysiological hallmark of PD, the modulation of the activity of STN neurons by DA has been the focus of many studies. STN neurons mainly express D2/3Rs and D5Rs on their membrane [116,117]. Most of our current knowledge about DA action on STN neurons comes from exogenous application of DA itself or agonists of the D1- and D2-like receptor families in acute brain slices. Application of low concentrations of DA or the D2-like receptor agonist quinpirole induces a depolarization of the membrane potential and an increase in the spontaneous firing rate of STN neurons [43,116,118]. This excitatory action of D2-like dopaminergic receptors is mediated by inhibition of Ca_V_2.2 (N-type) calcium channel currents by G_i/o_ βγ subunits. This inhibition reduces Ca_V_2.2 channel functional coupling with small-conductance Ca^2+^-dependent K+ (SK_Ca_) channels, promotes membrane depolarization, and increases firing discharge of STN neurons [116] (Figure 3A). On the other hand, D5R activation targets several conductances depending on the mode of discharge of STN neurons [119]. When STN neurons’ membrane potential is hyperpolarized, they fire bursts of APs and D5R activation potentiates Ca_V_1.3 calcium channel currents, prolonging burst duration [120] (Figure 3A). When STN neurons’ membrane potential is depolarized, these neurons fire single APs in a tonic fashion, and D5R increases their firing rate via the activation of cyclic nucleotide gated non-cationic channels [121]. Therefore, it has been proposed that D2Rs control the pattern of activity (tonic or burst) by setting the level of depolarization of STN, while D5Rs reinforce each mode of discharge [118].

### 4.2. Alteration of STN Autonomous Pacemaking in DA Depleted Rodents

It is widely accepted that the STN is hyperactive in PD and that STN neuron activity is composed of rhythmic and synchronous bursts of APs [122]. This pathological activity can have several origins, such as DA loss in the STN, leading to altered excitability of STN neurons or alterations in the BG network. Ex vivo studies suggested that the excitability of STN neurons is strongly downregulated in DD rodents [123,124], but the underlying mechanisms were unknown until recent work confirmed that STN neuron autonomous pacemaking is lost in neurotoxic and genetic mouse models of PD [125]. This loss of excitability seems to have a network origin, as it is triggered by the increased drive of iSPNs onto GP neurons, leading to disinhibition of STN neurons and excessive activation of NMDAR, which finally increases K_ATP_ channels [125]. Indeed, the expression of these channels has been reported to be increased after DD [126], which produces sufficient hyperpolarization of the membrane potential to silence STN neurons [125]. Another study suggests that DD alters the expression profile of HCN2 channels in the STN and also contributes to pathological activity in the BG in experimental PD [127].

### 4.3. Augmentation of Pallido-Subthalamic Transmission in Experimental Parkinson’s Disease Models

The GP provides the STN’s principal source of GABA. Single GP axons make sparse clusters of synaptic boutons distributed in distinct functional domains of the STN [128]. On average, a GP axon makes at least six synaptic contacts on the soma and the proximal dendrites of STN neurons. Thus, each GP neuron provides potent somatic inhibition, which is highly efficient to reset the autonomous activity of STN neurons [128]. GP neurons fire APs at a high rate [70,129] and this elevated activity influences, in an activity-dependent manner, the efficacy of GP-STN inputs that are characterized by STD [130,131]. GP-STN inputs are modulated by presynaptic D2/3Rs, which decrease the release probability of GABA and thus reduce the strength of this connection. It is also interesting to note that DA converts GP-STN synapses from low-pass filters to band-pass filters [131] favoring information transfer [132]. Under DD conditions, GP-STN GABAergic transmission is greatly augmented due to structural modifications of GP-STN synapses, which make more active zones per bouton [133] (Figure 3B). Recently, Chu and colleagues [134] have unraveled the mechanisms underlying this proliferation of GABAergic synapses. In this study, they showed that GABAergic long term potentiation (LTP) involves the excessive activation of NMDA receptors, which triggers 1) nitric oxide synthesis and release by STN neurons, which activates presynaptic protein kinase G (PKG) and increases GABA release probability, and 2) an increase in intracellular Ca^2+^ concentrations that activate a CaMKIIα pathway, which augments GABA_A_ receptor insertion on the post-synaptic side of GABAergic GP-STN synapses [134] (Figure 3B).

### 4.4. Loss of the Cortico-Subthalamic Pathway in Experimental Parkinson’s Disease Models

The cortico-subthalamic (Cx-STN) projection is well described anatomically. Cx-STN synapses are mainly found on the distal part of STN neurons’ dendrites [135,136]. The functional properties of Cx-STN glutamatergic synapses have been described recently. Around the resting potential of STN neurons, Cx-STN post-synaptic excitatory currents (EPSC) are mainly composed of AMPAR-mediated currents with a small NMDA component [134,137]. Interestingly, it has been shown that the strength of the Cx-STN inputs is under the control of DA and more specifically, of post-synaptic D5Rs which depress AMPAR-mediated EPSC through a PKA-dependent intracellular pathway [137]. Another study has also shown that pre-synaptic D2Rs modulate AMPA currents by decreasing glutamate release [138]. Together, these data suggest that Cx-STN synaptic transmission is modulated by DA, and hence, this pathway may be altered in PD. Indeed, it has been shown that, in DD rodents, Cx-STN innervation is reduced [139,140,141] but the mechanisms underlying this loss of Cx-STN glutamatergic inputs remain to be determined. It has been proposed that the alleviation of motor symptoms by M1 motor cortex deep brain stimulation (DBS) [142,143] relies on the restoration of Cx-STN functional connectivity of the remaining Cx-STN afferent inputs [141].

## 5. The Substantia Nigra Pars Reticulata

### 5.1. Neuronal Diversity in the Substantia Nigra Pars Reticulata

Together with the EPN, the SNr is one of the output nuclei in the basal ganglia. Like other BG nuclei, the SNr has been classically considered a homogeneous nucleus, and cell differences were based more on topographical innervation patterns than on neuronal diversity. Although it is well known that the main target of SNr is the thalamus [144], up to four neuron types have been identified based on the axonal projection targets. Type I cells project specifically to the thalamus, type II neurons target the thalamus, superior colliculus and pedundulopontine tegmental nucleus, type III cells project to the periaqueductal gray matter and thalamus, and type IV neurons send projections to the deep mesencephalic nucleus and the superior colliculus [145,146]. In addition to this long axonal arborization, SNr neurons are also highly collateralized [145,147], and most neurons receive robust inhibitory synaptic inputs even in the presence of strong activation [148,149].

In the SNr, GABAergic projection neurons are the largest cell population, but discrete clusters of dopaminergic neurons are also present in the caudomedial region of the nucleus [150]. The majority of GABAergic neurons in the SNr express PV [151,152] and a small subset of cells express calretinin [153,154], nitric oxide synthase, or acetylcholine transferase [150,155]. These cells differ in their neurochemical content and their topographical and morphological profiles. Thus, in the rostrolateral SNr, GABAergic cells are large and contain PV and nitric oxide synthase. In caudomedial positions, most of them are small and express only PV, and in rostromedial portions, they are predominantly small and contain either calretinin, nitric oxide synthase or PV [150]. As for receptor expression, the vast majority of PV-positive neurons also display GABA_A_ receptor α1 subunits and at least one (GABA_B_R2) of the heteromeric subunits of the GABA_B_ receptor [156,157]. Moreover, cells with colocalized PV and calbindin also express purinergic receptor P2 [158].

Neurochemical analysis clearly distinguishes three different classes of cellular content, but electrophysiological characterization of GABAergic cells and correlation with the neurochemical profile has yet to be fully developed. PV-positive cells have been associated with high firing frequency [159], but this characteristic does not seem to be cell-specific. Indeed, calretinin-containing cells show similar projection targets, local arborization, morphological, and electrophysiological characteristics to PV-positive cells [160]. More recent studies describe four subtypes of GABAergic cells in the SNr, whose electrophysiological profile is complex and varies according to the posture and movement of the animal [161,162].

### 5.2. Ionic Conductances Underlying Pacemaking in Snr Neurons

Like other ESNs, SNr neurons are autonomously active. This regular spiking conveys an efficient tonic inhibitory drive onto motor thalamic nuclei [163,164,165]. Repetitive AP generation in SNr neurons is primarily supported by the subthreshold slowly inactivating Na_V_ channels, while SK channels are critical to maintain the precision of autonomous pacemaking in these neurons [18]. HCN are also present in SNr cells but are not activated in the range of potential associated with pacemaking activity. It has also been shown that leak conductances are critical for the maintenance of SNr neurons at depolarized potential [166,167]. A molecular and electrophysiological study suggested that transient receptor potential channel family (TRPC3) channels are expressed in SNr neurons and contribute to their depolarized resting potential [167]. However, another study showed that spontaneous firing of SNr neurons is unaffected after genetic deletion of these channels [166], questioning the participation of these channels in the pacemaking of SNr neurons. This discrepancy can be explained by the lack of a specific blocker of non-specific cationic channels used by Zhou and colleagues to show the involvement of TRPC3 channels in the depolarized potential of SNr neurons. In addition, a more recent study has discovered that SNr neurons express the sodium leak channel, NALCN, and that genetic deletion of NALCN impairs spontaneous firing in these neurons [168]. Because the activity of NALCN is strongly dependent upon glycolysis, alterations in this metabolic pathway can significantly impair autonomous pacemaking in SNr neurons [166,168] and contributes to pathological activity of this nucleus. As with other ESNs, Kv3-channels are essential for SNr neurons to fire APs at high frequency, as they provide rapid repolarization of the membrane potential following AP depolarization [169]. An extensive description of the ionic mechanisms governing SNr neurons excitability can be found in the following review [170].

### 5.3. Dopamine Modulation of Intrinsic Excitability of Snr Neurons

Dopaminergic modulation of SNr is achieved through an unconventional release of DA by the dendrites of SNc neurons (for review, see [171]), which constitutes an ultra-short dopaminergic pathway. In situ hybridization and immunohistochemistry studies have shown that several DARs are expressed in the SNr. The most intense labeling is for D1Rs, especially in striato-nigral terminals. D4Rs and D5Rs are present in the SNr but mainly on perikarya [172,173,174,175,176], and D1/D5Rs have been found both on SNr neurons and astrocytes [177]. D1-like agonists excite SNr neurons, and this modulation is mediated by a PKA-dependent enhancement of the constitutively active TRPC3 channels, which depolarizes SNr neurons [178]. Interestingly, this effect was mimicked by artificially elevating ambient DA levels, which supports its physiological and functional relevance [178]. In addition, acute blockade of D1-like and D2-like receptors induces hyperpolarization of SNr neurons and a switch from tonic regular firing to irregular or burst firing [47]. This pharmacological manipulation resembles the activity of SNr neurons recorded in vivo in anesthetized [179,180,181,182] and awake [183] DD rodents. The cellular mechanisms underlying the changes in rate and pattern of SNr under DD have not been explored ex vivo. It will be of great interest to further investigate which intrinsic conductances of SNr neurons are directly impacted by the loss of DA.

### 5.4. Alteration of Gabaergic and Glutamatergic Transmission in Snr Neurons in Experimental Parkinson’s Disease Models

Anatomical studies have shown that single SNr neurons receive convergent afferents from dSPNs, the GP and the STN (for review [184]), supporting a strong integrating function of this output nucleus of the BG. Like in the GP, pallidal and striatal GABAergic inputs make symmetric synapses with the soma/proximal dendrites and distal dendrites of SNr neurons, respectively [185,186,187]. In addition to their subcellular locations, these two GABAergic synapses have also distinct properties. Striato-nigral (STR-SNr) IPSCs exhibit STF, while pallido-nigral (GP-SNr) synapses display STD [188]. These two synapses are also different regarding the pre-synaptic control exerted by DAR. GP-SNr synaptic transmission is reduced by D4R activation [189,190] while STR-SNr GABA release is augmented by D1Rs [189,191,192,193]. Because of its opposite action on STR-SNr and GP-SNr synapses, DA maintains a certain equilibrium between somatic and dendritic inhibitory inputs received by SNr neurons, whose precise function remains to be elucidated. The impact of DD on GP-SNr synaptic transmission is unknown but has been investigated for STR-SNr synapses [194]. In their study, the authors report that DD induces a strong increase in STR-SNr IPSC amplitude and show that dysfunctional GABA_B_ receptors and loss of presynaptic reduction of GABA release probability was responsible for this augmented transmission [194].

The STN is the main provider of excitatory inputs to the SNr, and the properties of STN-SNr synapses have been well characterized. Electrical stimulation of STN axons triggers monosynaptic EPSCs [195,196], while the same type of stimulation delivered in the STN itself gives rise to complex EPSCs [197,198], which are believed to be generated by the activation of STN local axon collaterals [197,199]. Both D1Rs and D2Rs are present on STN-SNr synaptic terminals and activation of D1Rs enhances while activation of D2Rs decreases STN-SNr EPSC amplitude [196]. On the other hand, complex EPSCs which have been shown to promote burst-firing in SNr neurons [197,200] are reduced by the activation of D2-like receptors [200]. Only long-term depression (LTD) has been reported at STN-SNr. The induction of LTD requires post-synaptic D1R activation and is expressed through NMDAR-dependent endocytosis of AMPARs, which depresses EPSC amplitude by almost 50% [195]. This mechanism represents a specific feature of this synapse compared to cortico-striatal synapses [201], but like in the STR, DD completely abolishes STN-SNr LTD [195], supporting the hypothesis that increased synaptic transmission at STN-SNr synapses contributes to pathological activity of SNr neurons. DA-glutamate interplay dysfunctions at the post-synaptic density have been suggested as critical determinants of major psychiatric disorders [202,203] and could be emphasized as potential molecular targets at STN-SNr synapses to restore a normal level of activity of SNr neurons.

Overall, there is compelling evidence suggesting that both STR-SNr and STN-SNr synaptic transmission strength is regulated by DA and is pathologically-enhanced in experimental PD.

## 6. The Entopeduncular Nucleus

### 6.1. Anatomical Organization and Cellular Diversity in the EPN

The EPN is, in addition to the SNr, the other output nucleus of the BG. Early anatomical studies have suggested that the EPN is subdivided in rostral and caudal portions, with the former being enriched in somatostatin-positive (SOM+) neurons and the latter being composed of PV-expressing neurons [151,204,205]. More recently, a third neuronal population, negative for both SOM and PV, has been described [206]. The proportion of these three populations has been estimated as 28% (PV+/SOM-), 46% (PV-/SOM+), and 25% (PV-/SOM-). Additional molecular profiling has confirmed the existence of these three neuronal subtypes and has specified their molecular identity [207]. It has also been shown that the EPN is organized in a PV-rich core and a PV-poor shell fashion [206]. Retrograde tracing studies have demonstrated that SOM+ EPN neurons project to the lateral habenula (Lhb) [208,209], while PV+ EPN neurons innervate the motor thalamus [151,208]. Furthermore, Lhb- and thalamic-projecting EPN neurons are differentially innervated by subclasses of pallidal and striatal neurons [207,208]. EPN-thalamic neurons are innervated by PV+ GP neurons and matrix SPNs, while EPN-Lhb neurons receive inputs from PV- GP neurons and striosome SPNs, suggesting that these two EPN cell types are part of distinct functional networks. Indeed, it has been proposed that thalamic-projecting EPN neurons control motor program selection, while Lhb-projecting EPN neurons participate in the evaluation of the motor outcome [208].

### 6.2. Autonomous Pacemaking in EPN Neurons.

EPN neurons fire at a high rate in vivo both in anesthetized and awake rodents [210]. Like other ESN, it is likely that this elevated discharge relies on intrinsic properties of EPN neurons. Indeed, EPN neurons are spontaneously active ex vivo [211], but not much is known about the properties of the channels responsible for autonomous pacemaking in these neurons, and DA modulation of their excitability has never been investigated. Based on their firing pattern and properties ex vivo, two types of EPN neurons have been described [211,212]. Type I EPN neurons are spontaneously active, generate a sag upon membrane hyperpolarization, and generate a rebound-burst when hyperpolarizing current injection ends, suggesting that they express HCN and CaV3-type calcium channels, respectively. Type II EPN neurons are not autonomously active. They have low HCN channel expression and potassium A-type currents, which generate a slow depolarizing ramp at the end of hyperpolarizing pulses of currents [211]. Interestingly, most type I, but not type II, EPN neurons exhibit GABAergic IPSCs upon electrical stimulation of the striatum [212], which suggests differential innervation of these two cell types. Molecular and electrophysiological correlative studies would be required to determine if Lhb-projecting and thalamic-projecting EPN neurons possess specific electrophysiological signatures. To our knowledge, neither the modulation exerted by DA nor the direct consequences of DD on the excitability of EPN neurons have yet been investigated.

### 6.3. Dopamine Modulation of Gabaergic and Glutamatergic Transmission in the EPN

Like the SNr, the EPN receive convergent inputs from the striatal direct-pathway, the GP, and the STN [185,213]. Pallido-entopeduncular (GP-EPN) and striato-endopeduncular (STR-EPN) synapses display the same properties as STR-SNr and GP-SNr synapses, i.e., STF and STD, respectively [214].

All DAR subtypes have been found in the EPN [215,216]. Their presynaptic distribution appears to be very similar to that of the SNr: D2-like receptors are present on GP-EPN terminals, while D1Rs are found on STR-EPN synapses [215,216]. The modulation exerted by DA on these two pathways seems to also follow the principles described in the SNr, as D2-like receptors depress GP-EPN transmission whereas STR-EPN inputs are potentiated by D1Rs [215,217]. The impact of DD on these synapses has not been tested, but one may speculate that DD will facilitate GP-EPN synapses and depress the STR-EPN synapses, therefore generating an imbalance between direct and indirect pathway GABAergic inputs in the EPN.

The STN sends glutamatergic projections to the EPN [218]. The STN-EPN synapses display spike-timing dependent LTD [219], which, like in the SNr, can be viewed as an adaptive mechanism to regulate the impact of the STN on the output of the BG.

## 7. Consequences of Cellular and Synaptic Dysfunctions for Abnormal Neural Dynamics in the Basal Ganglia during Parkinson’s Disease

One critical aspect that remains to be addressed is defining the link between the different molecular/synaptic alterations occurring after DA loss and the pathological BG activity recorded at the network level. This lack of knowledge raises important questions that need to be assessed to better understand how BG activity becomes dysfunctional in the DD state. In particular, it seems critical to define if the synaptic modifications are the cause of the neuronal network dysfunction present in PD or if it is the other way around. Also, being able to determine the respective contributions of the synaptic vs. the network alterations to Parkinson’s motor symptoms might help in designing new therapeutic strategies to target more precisely the pathophysiological process in PD. These are not trivial questions to tackle, especially considering the difficulty of determining if these changes are truly pathogenic, compensatory, or by-products of other (yet to be defined) neuronal changes. Another challenge facing the field is the fact that the network abnormalities underlying the motor disturbances of PD are still enigmatic. Indeed, while traditional views on the pathophysiological organization of BG circuits have highlighted the contribution of firing rate alterations to explain the motor impairments present in PD [8,220], other neuronal activities, such as changes in the firing pattern [182,221,222,223,224], or increases in neuronal synchronization [77,78,225,226], have also been associated with the Parkinson’s disease condition [227,228,229]. Whether these changes underlie the pathophysiology of PD is not known, but their presence is positively correlated with the PD bradykinesia and rigidity score [230] and can be used to distinguish the PD state [231]. Here, we will briefly review the principal neuronal network changes occurring after DD, the arguments in favor or against their causative contribution to the pathophysiology of PD, as well as how these network changes might be affected by the synaptic alterations previously described in this review.

First, considering the firing rate modification, it has been shown that the loss of DA creates an imbalance in activity in striatal projection neurons that leads to iSPN hyperactivity and dSPN hypoactivity [73,232,233,234]. This increase in iSPN activity induces a cascade of firing rate changes along the indirect pathway: notably decreased activity of GP neurons [14,71,235,236] and increased activity of STN neurons [221,237]. These modifications lead to increased firing of the BG output nuclei SNr/EPN [224,238], which translate into over-inhibition of the thalamo-cortical motor circuits [239]. The most compelling evidence in favor of a pathogenic contribution of the firing rate alterations comes from studies that have used cell-type-specific optogenetic [9,240,241] or pharmacogenetics [134,242] manipulations to reproduce the firing rate changes present in PD. In particular, specific excitation of iSPNs induced a reduction of locomotion mimicking the PD akinetic state [9,134,242], whereas activation of the dSPNs in PD model mice reduced the motor deficit and elicited locomotion [9]. In addition, a direct link between the increase in SNr output activity and motor suppression has been established consequently to iSPN opto-stimulation [240], thus supporting the view that BG outputs exert an inhibitory influence on movement control [146]. The timing at which the firing rate of striatal neurons is modified after DA loss is not known, but it is likely to be fast. Indeed, the expression of the messenger RNA for *c-fos*, a marker of neuronal reactivity often used as a correlate of neuronal activity [243], is increased in iSPNs 75 min after 6-OHDA injection. This timing parallels the DA loss [244]. Therefore, it seems reasonable to expect that the firing rate modifications along the indirect pathway are occurring before the synaptic reorganization (which might appear with longer time-scale after DA loss). This assumption favors the idea that some of the synaptic changes observed in the DD state might thus be corrective by nature, compensating for the abnormal firing rate activities established in the indirect pathway. With this in mind, the selective loss of glutamatergic synapses in iSPNs [245], the increased connectivity of striatal fast-spiking interneurons onto iSPNs [246] and the increased number [133] and strength of GP synapses onto STN neurons [133,134] might represent global homeostatic changes of the indirect pathway aiming to counterbalance the increased activity of iSPNs. Similarly, the strong synaptic depression of the cortico-STN excitatory inputs present both in rodents and monkey PD models [139,140] could also be an adaptive change to limit STN hyperactivity and, at the end-stage, the negative influence of BG output onto thalamo-cortical and brainstem circuits. The apparent discrepancy between the functional restoration of the Cx-STN pathway by motor cortex DBS (see Section 4.4) and an adaptive reduction of this pathway to counteract STN hyperactivity can be explain if motor cortex DBS acts by breaking pathological activity and promoting corticofugal [247] information transfer rather than simply increasing Cx-STN synaptic transmission. However, despite all the experimental evidence arguing for an important contribution of BG overall firing rate modification in experimental PD, it is still unknown if they represent the sole pathogenic cause of Parkinsonism [248,249,250]. Additionally, many experimental findings have directly challenged the predictions of the rate model. For example, while recordings in MPTP-treated monkeys have clearly established STN firing hyperactivity [221,235], changes in striatal firing activity as described by the rate model have been more conflicting. Indeed, while one study described a profound increase in striatal neuronal firing [251], another reported no change [235]. In addition, recordings in DD mice have shown a reduction in the autonomous properties of STN neurons which, when restored using chemogenetic excitation of STN, improves motor dysfunction in DD mice [125]. Considering the fact that motor recovery in PD is classically obtained using suppression [252] or reduction [238] of STN neuronal firing, these results were totally unexpected, and in contrast with the motor-suppressing effect predicted by the rate model when STN increase their activity. Another piece of experimental evidence that challenges the rate model is the fact that SNr firing output in DD mice is hypoactive and not hyperactive [183]. Taken together, the direction of the firing rate changes in various nodes of BG circuits might not be consistently present across different animal models of PD (i.e., rats vs. mice vs. monkeys), thus raising the possibility that other neuronal changes might more reliably contribute to the motor dysfunction of PD.

Abnormal expression of synchronized oscillatory activity in the beta frequency band (12-35 Hz) has been one of the most prominent neuronal changes consistently detected in PD patients and has been related to akinesia/bradykinesia syndromes in PD [253,254]. Accordingly, a correlation between the level of beta oscillation expression and the motor deficit has been described in the off-medicated state of PD [255] (but see [256]). In addition, imposing abnormal beta synchronization through external stimulation at beta frequency slows down movements both in healthy [257] and PD subjects [258], which suggests a pathogenic role (although this stimulation protocol likely altered the level of neuronal firing, which was not accounted for in the motor perturbation). What are the neuronal circuits generating this abnormal beta synchronization? In theory, any network with delayed negative feedback properties can generate oscillatory activity [259], and because the organization of the BG is principally composed of parallel feedback loops, many BG circuits could potentially generate beta oscillations [260,261,262]. It was first proposed that the abnormal BG synchronization is an emergent property of the GP-STN network [263], but this work was performed in organotypic culture and so far, there is no in vivo evidence that the GP-STN network could sustain synchronized rhythmic activity without the cortex [222,264]. Other proposed circuit generators of beta oscillations are the cortex [265], the striatum [266], the STR-GP network [267], and the hyperdirect loop [268]. One influential hypothesis suggests that beta oscillations are generated at the cortex at physiological levels, which propagate in BG circuits through the hyperdirect pathway where they are abnormally amplified by the reciprocally-connected GP-STN microcircuits [265,269,270]. There is indeed a strong rationale to suggest that the GP-STN network is involved in the maintenance and the propagation of oscillatory activity both in vivo and ex vivo [13,14,71,271,272]. Nonetheless, the specific contribution of these different circuit elements has never been tested with optogenetic tools that are reversible and offer high temporal resolution. One aspect to consider is the timing at which abnormal beta synchronization is generated after the DA loss, as it reveals important features regarding its contribution and interaction with the firing rate or synaptic modifications. Interestingly, beta oscillations are not detected in PD rat models until >4 days after the lesion and reach a plateau of expression after 10 days. This slow time-scale of generation contrasts with the fast motor deficits that appear within hours following the loss or the blockage of DA transmission [237,273,274]. Similarly, monkeys and rats present Parkinson’s associated motor symptoms following chronic MPTP or 6-OHDA treatment before the emergence of BG synchronous oscillatory activity [250,275]. Taken together, this suggests that beta synchronized oscillations might arise through long-term adaptive changes that are slower than the changes in firing rate, but which could parallel the timing of the synaptic alterations. This potentially highlights the relationship between these two kinds of activity, but whether one is the consequence of the other is currently not known. Similarly, the link between the firing rate modifications and the change in neuronal synchronization has never been investigated. We propose here the hypothesis that the GP firing rate alterations in PD trigger maladaptive modifications that favor GP neuronal synchronization. Indeed, in a normal animal, the firing of GP neurons is principally governed by their autonomous pacemaker properties, which are different across neurons due to variability in channel density [53,79]. Active decorrelation mechanisms caused by GP neurons’ intrinsic properties and mutual inhibition through axon collaterals prevent neuronal synchronization in normal conditions [226,276]. This indicates that the intra-GP GABAergic control in normal conditions is mainly achieved by GP neurons. In contrast, the loss of DA in PD, which causes the increase in iSPN firing, operates a switch in the GP GABAergic control from intrinsic (i.e., GP) to extrinsic (i.e., STR) sources. This change of GABA transmission is certainly at the origin of maladaptive changes at the molecular and synaptic level that favor the generation of synchronized oscillatory activity in GP neurons. Because of the critical dichotomous organization of GP neurons into prototypic and arkypallidal neurons [14], this abnormal oscillatory activity can then propagate to the whole BG circuit. In this scheme, the abnormal network synchronization is caused by maladaptive changes occurring at the cellular and synaptic levels and is a consequence of the firing rate perturbation in the indirect pathway. It is important to mention though that this abnormal beta synchronization has been detected in PD animal models, such as 6-OHDA-lesioned rats [237,247,277,278,279] and MPTP-treated monkeys [235,280], but not in 6-OHDA or α-synuclein mice [183,281], which raises the question of their contribution to the pathophysiology of PD. Indeed, future studies will have to precisely define the specific behavioural contributions of the different neuronal activities and how they impact the motor symptoms of PD.

STN or GPi DBS is highly effective to alleviate PD motor symptoms and it is widely thought that DBS achieves its therapeutic action by disrupting pathological oscillatory activity in the motor cortex/BG loops [282,283]. Considering the complex anatomical connectivity of the BG and the possible concomitant local and distant effects of STN-DBS [247,284,285,286], the mechanisms through which high frequency stimulation breaks down pathological activity and restores motor function remains highly debated. Because axons are known to be more responsive to electrical stimulation than cell bodies [287] and that globally ESN synapses are pathologically-enhanced in PD models one may speculate that STN-DBS has a stronger impact on synaptic transmission and neurotransmitter release rather than on neuronal excitability, which allow them to re-establish information transfer within the BG and execution of motor programs. In addition to what has been done already [142], the future development of optogenitically-based DBS protocols will certainly help to better understand where in the network and by which precise mechanism(s) STN-DBS restores motor function in PD.

## 8. Concluding Remarks

In this review of the literature, we have highlighted that every ESN of the BG receives a functional DA innervation that controls both the post-synaptic excitability as well as the strength of their afferent synaptic inputs. In PD, extrastriatal hypodopaminergy provoked by the loss of SNc neurons must participate, in addition to the alterations described in the striatum, in the pathophysiological activity of the BG network. According to all the evidence accumulated over the past decades, we propose that DA loss induces an imbalance between ESN neuron intrinsic excitability and ESN synapses, in favor of the latter. Autonomous pacemaking is reduced in all ESN while synaptic transmission is enhanced, leading to a switch from intrinsically-driven oscillatory activity in ESN to a synaptically-driven pattern of activity, which promotes hypersynchronous oscillatory activity in the BG network in PD (Figure 4B).

## Figures and Tables

**Figure 1 cells-08-01005-f001:**
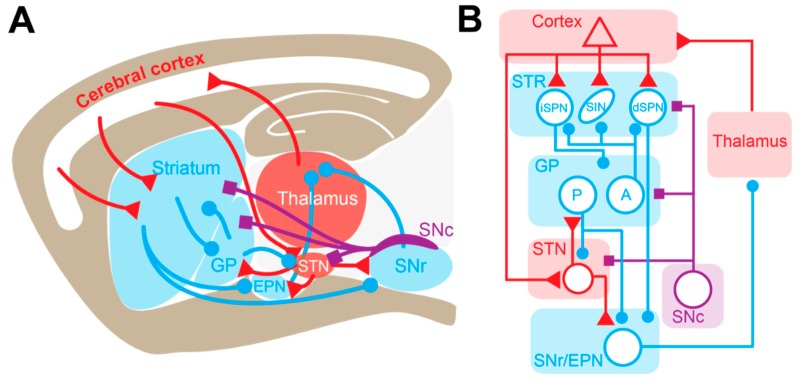
Basal ganglia circuitry in rodents. (**A**) Schematic representing the main connections of the basal ganglia network in a sagittal section of the rodent brain. (**B**) Corresponding anatomical-functional diagram adapted from the Albin model of the basal ganglia [8]. Dopaminergic, GABAergic, and glutamatergic projections and nuclei are depicted in blue, purple and red, respectively. A: arkypallidal neuron; dSPN: direct-pathway spiny projection neuron; EPN: entopedoncular nucleus; GP: globus pallidus; iSPN: indirect-pathway spiny projection neuron; P: prototypic neuron; SNc: substantia nigra pars compacta; SNr: Substantia nigra pars reticulata; SIN: Striatal interneurons; STN: subthalamic nucleus.

**Figure 2 cells-08-01005-f002:**
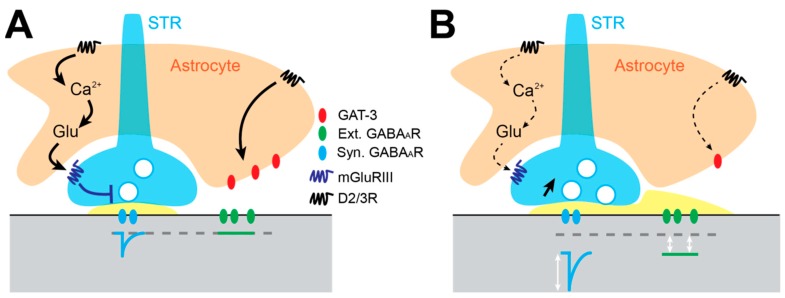
Mechanisms underlying GABAergic synaptic alterations in the GP in experimental PD. (**A**) In DA-intact conditions, D2 receptors present on astrocyte membranes elevate intracellular calcium levels and promote glutamate gliotransmission. Glutamate activates group III mGluR (GluRIII), which reduces GABA release probability at striato-pallidal synapses. D2Rs also regulate GAT-3 activity by an unknown mechanism, preventing GABA spillover from GABAergic synapses and hence activation of extrasynaptic GABA_A_ receptors. (**B**) Under DA-depleted conditions, D2Rs are no longer activated, glutamate gliotransmission is reduced and mGluRIII-dependent reduction of GABA release is lost, leading to an increase in striato-pallidal transmission. The loss of D2R modulation of GAT-3 triggers a downregulation of the expression of the transporters, and the elevation of extracellular GABA concentrations leads to the activation of extrasynaptic GABA_A_ receptors, which favors tonic inhibition.

**Figure 3 cells-08-01005-f003:**
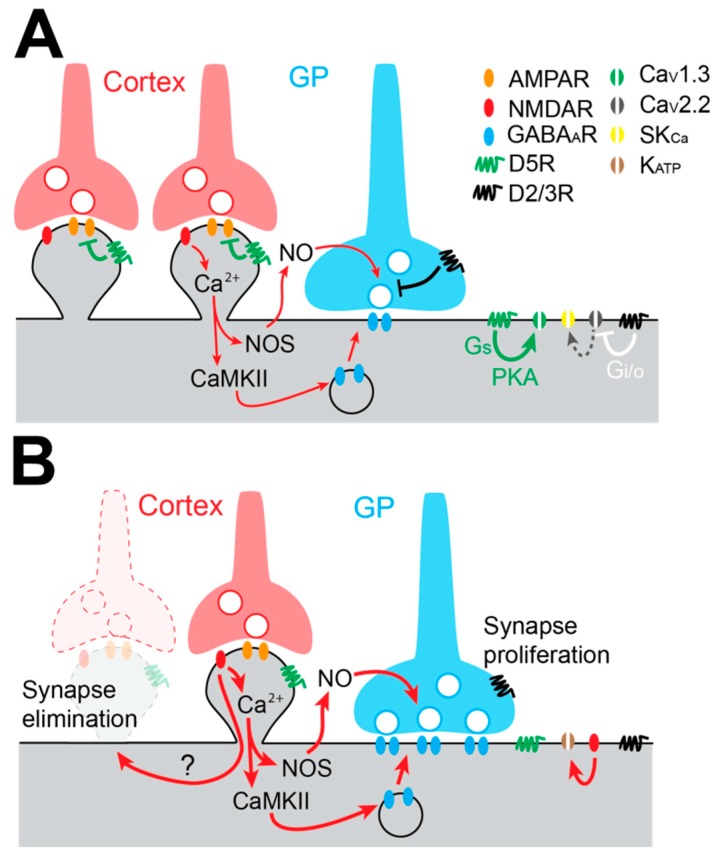
Cellular and synaptic alterations in the STN in experimental PD. (**A**) Schematic of the main post-synaptic and pre-synaptic molecular pathways involved in DA-dependent cellular excitability and synaptic plasticity. (**B**) Under DD conditions, post-synaptic and pre-synaptic DA modulation is lost. This triggers NMDA-dependent M1 cortex synapse pruning, NMDA-dependent and NO-dependent heterosynaptic LTP (synapse proliferation) at pallido-subthalamic GABAergic synapses and NMDA-dependent activation of K_ATP_ channels, which reduces STN autonomous pacemaking.

**Figure 4 cells-08-01005-f004:**
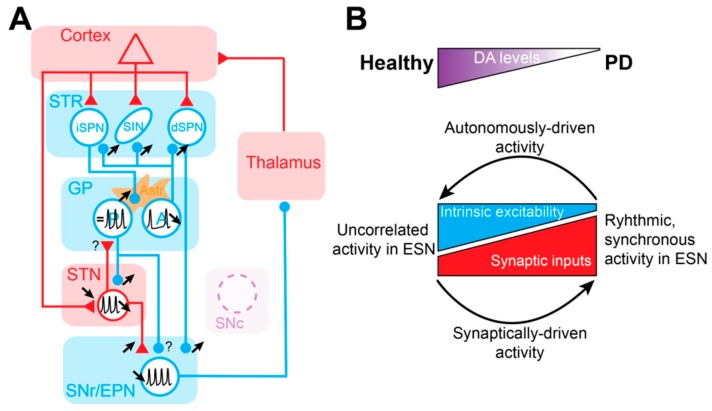
Cellular and synaptic alterations in the ESN in experimental PD. (**A**) Summary diagram of the main modifications in cellular excitability and synaptic transmission observed in the BG network under DA-depleted conditions. Black arrows indicate increase or decrease in synaptic transmission or intrinsic excitability (represented by black APs). Astr.: astrocyte involvement in GABAergic dysfunction in GP. GABAergic and glutamatergic projections and nuclei are depicted in blue and red, respectively. Loss of SNc DA neurons is represented by the dashed purple circle. (**B**) Schematic illustrating the modification in the balance between intrinsic neuronal excitability and synaptic strength in ESN as a function of the levels of DA.

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
