# Peer review of "Cellular and Synaptic Dysfunctions in Parkinson’s Disease: Stepping Out of the Striatum"

_cells, 2019, doi:10.3390/cells8091005_

Round 1

Reviewer 1 Report

This is an authoritative and comprehensive review of basal ganglia circuitry with an emphasis on extra-striatal connections. It provides a well written, informative and important perspective on the normal functioning of this circuitry and the dysfunction occurring in Parkinson’s disease.

The paper begins with the basal ganglia circuitry shown in Figure 1B and presents evidence interpreted in terms of this circuitry and the elaboration shown in Figure 4A. This is a very clear and logical approach, but it would help the reader to have a more elaborate description of both figures in the figure legends. For example, it appears that neurons are color-coded (blue = GABA, red = glutamate, purple = dopamine), but this is not explicitly stated. In Figure 1B, does the GP circle on the left refer to prototypic neurons and that on the right to arkypallidal neurons? These could be labeled too. The review included a nice discussion of astrocytes in the globus pallidus and their role in dopamine signaling. Does the orange star shape in Figure 4A indicate this astrocyte involvement? This is not indicated anywhere on the figure or in the figure legend. Given the clinical significance of deep-brain stimulation, the authors might add some discussion about how it affects the basal ganglia circuitry. In lines 517-520, the authors mention that the reduced cortico-STN inputs could be an adaptive mechanism, while in lines 313-315, they mention that DBS relies on the restoration of these inputs. Could the authors elaborate or discuss this discrepancy?

Some minor corrections should be addressed:

Lines 4-5: The names of the authors appear to be inverted (surname first, given name last). This should be checked and corrected. The label for the thalamus is difficult to make out in both Figure 1B and Figure 4A. Using a darker shade for the thalamus would make it much easier to read. In Figure 1B, the nucleus on the bottom is labeled “SNr/EPN” but in Figure 4A it is simply labeled “SNr.” Is this an oversight or was EPN omitted for a reason? Line 121: “show of a” should read “show a” Line 123: “neuron” should be plural (neurons) Line 127-128: should this read “TF FoxP2 and a low”? Line 249: “mediated inhibition” should read “mediated by inhibition” Line 323: “type II” should be ‘type III’ Line 327: “present of” should read “presence of” Line 336: “receptors” should read “receptor” Line 515: “increased in the number and strength” should read “increased number and strength” Line 583: “propagates” should read “propagate”

Reviewer 2 Report

In the present review, the functional relevance of dopamine modulation outside the striatum in both normal and pathological conditions will be discussed. Overall, I found the present review very interesting and scientifically sound. However, I have some minor concerns on it:

1) In the introduction I suggest the Authors to more clearly report the aims and the scope of the review.

2) Moreover, the Introduction should be splitted in two separate parts (from line 30 to 46 and from 47 to 66).

3) I suggest the Authors to add the methods employed to conduct the review in a separate paragraph.

4) I suggest the Authors to add a brief comment on DA and GLU interplay in the post-synaptic density with appropriate references (see Tomasetti et al. Int J Mol Sci. 2017 Jan 12;18(1) and De Berardis et al. Int J Mol Sci. 2018 Sep 23;19(10).)
